# ProofNet: Autoformalizing and Formally Proving Undergraduate-Level Mathematics

## Abstract

We introduce ProofNet, a benchmark for autoformalization and formal proving of undergraduate-level mathematics.[1] The ProofNet benchmarks consists of 371 examples, each consisting of a formal theorem statement in Lean 3, a natural language theorem statement, and a natural language proof. The problems are primarily drawn from popular undergraduate pure mathematics textbooks and cover topics such as real and complex analysis, linear algebra, abstract algebra, and topology. We intend for ProofNet to be a challenging benchmark that will drive progress in autoformalization and automatic theorem proving. We report baseline results on statement autoformalization via in-context learning. Moreover, we demonstrate improvements over our baselines by applying *prompt retrieval* and *distilled backtranslation* techniques.

## 1 Introduction

The creation of an automatic mathematician, that is, a system capable of autonomously posing conjectures and proving theorems, is a longstanding challenge in mathematics and artificial intelligence (Gelernter, 1959). In recent years, neural generative language modeling has emerged as a promising approach to automating aspects of mathematics (Rabe and Szegedy, 2021).

One approach to applying language models to mathematics has been to treat mathematical reasoning in natural language as a sequence learning task (Welleck et al., 2021a; 2022; Lewkowycz et al., 2022). A key advantage of mathematical reasoning in natural language is the abundance of natural language mathematics data on the internet (Lewkowycz et al., 2022).

An alternative approach is to use language models to guide formal proof-search in an interactive theorem prover (ITP) (Whalen, 2016; Yang and Deng, 2019; Wang and Deng, 2020; Polu et al., 2022; Jiang et al., 2022a; Lample et al., 2022; First et al., 2023). A salient advantage of this method is that the ITP acts as a verifier for the language model's reasoning, enabling the natural implementation of bootstrapping techniques such as expert iteration (Silver et al., 2017; Polu et al., 2022; Lample et al., 2022).

*Autoformalization*, the task of automatically formalizing mathematics, seeks to build a bridge between informal and formal mathematical reasoning (Wang et al., 2018; Szegedy, 2020; Wu et al., 2022a; Jiang et al., 2023), with the potential of extracting a training signal from vast corpora of natural language mathematics data while still grounding a system's reasoning in verified formal logic. However, the small amount and low diversity of parallel data between informal and formal mathematics means that autoformalization suffers from a lack of standard benchmarks to guide progress in the field.

To remedy this gap, we propose ProofNet, a benchmark consisting of parallel natural language and formal mathematics that can be used to evaluate autoformalization and theorem proving. The ProofNet benchmark consists of 371 parallel formal theorem statements, natural language theorem statements, and natural language proofs sourced from the exercises of popular undergraduate-level pure mathematics textbooks. Formal statements are expressed in the Lean 3 theorem prover (de Moura et al., 2015), and depend on Lean's mathlib (mathlib Community, 2020).

---

[1] Data provided within the supplementary materials

### Lean mathlib

**Formal theorem statement:**

```
theorem
exists_subgroup_card_pow_prime
  [fintype G] (p :  ℕ)
  {n :  ℕ} [fact p.prime]
  (hdvd:  p^n | card G) :
  ∃ K : subgroup G,
  fintype.card K = p^n
```

### ProofNet dataset (ours)

**Formal theorem statement:**

```
theorem exercise_4_5_14
  {G : Type*} [group G]
  [fintype G]
  (hG :  card G = 312) :
  ∃ (p :  ℕ) (P : sylow p G),
  P.normal
```

**Natural language theorem statement:**
Prove that a group of order 312 has a normal Sylow $p$-subgroup for some prime $p$ dividing its order.

**Natural language proof:**

*Proof.* Let $n_{13}$ be the number of Sylow 13-subgroups of $G$. Then by Sylow's Theorem, $n_{13} \equiv 1 \mod 13$ and $n_{13}$ divides $2^3 \cdot 3 = 24$. This implies $n_{13} = 1$, so that there is only one Sylow 13-subgroup, which is consequently normal. The last assertion follows from the fact conjugation preserves the order of a subgroup. So if there is only one subgroup $H$ of order 13, then for any $g \in G$, we have $\left|gHg^{-1}\right| = |H| = 13$, so $gHg^{-1} = H$, i.e., $H$ is normal. $\qquad\square$

Figure 1: A sample theorem statement from mathlib, show on the left, and a sample theorem statement from ProofNet, shown on the right. mathlib emphasizes including the most abstract and general formulations of mathematical results, whereas ProofNet predominantly tests the ability of models to apply those results to concrete problems.

Language-model-based theorem provers and autoformalization systems have typically been evaluated on benchmarks consisting of competition and olympiad-style problems (Zheng et al., 2022; Wu et al., 2022a). While such problems require complex reasoning, their solutions only depend on a relatively small set of elementary facts about integers, real numbers, counting, and geometry. In contrast, modern research mathematics requires the mastery of a massive body of theory made up of thousands of definitions, lemmas, and theorems. The Lean 3 formalization of perfectoid spaces, an important definition in research-level arithmetic geometry, depends on over 3000 distinct theorems and definitions (Buzzard et al., 2020). How to effectively reason over such a large repository of knowledge is an important unsolved problem in applying language models to mathematics (Irving et al., 2016; Wu et al., 2022b; Tworkowski et al., 2022) .

ProofNet falls short of requiring mastery of all of modern mathematics, but poses the still ambitious goal of reasoning over the core of an undergraduate mathematics, including basic analysis, algebra, number theory, and topology. We hope that this benchmark will spur the development of language models that are able to reason effectively over large knowledge bases.

In order to obtain stronger baselines on ProofNet, we train and open-source the PROOFGPT language models at scales of 1.3 billion and 6.7 billion parameters. These models are trained on the proof-pile, an 8.3 billion token dataset of mathematical text. To our knowledge, these are the only open-source language models fine-tuned for general mathematics.

We establish baselines for ProofNet theorem autoformalization using in-context learning (Brown et al., 2020). Moreover, we introduce two novel theorem autoformalization methods that outperform our few-shot baselines. *Prompt retrieval* uses nearest-neighbor search against an embedding database to create a prompt consisting of the mathlib declarations most relevant to a particular natural language theorem. *Distilled backtranslation* is a method inspired by work in unsupervised machine translation (Lample et al., 2017; Han et al., 2021a) that finetunes a language model for autoformalization at a large scale without the need for parallel data.

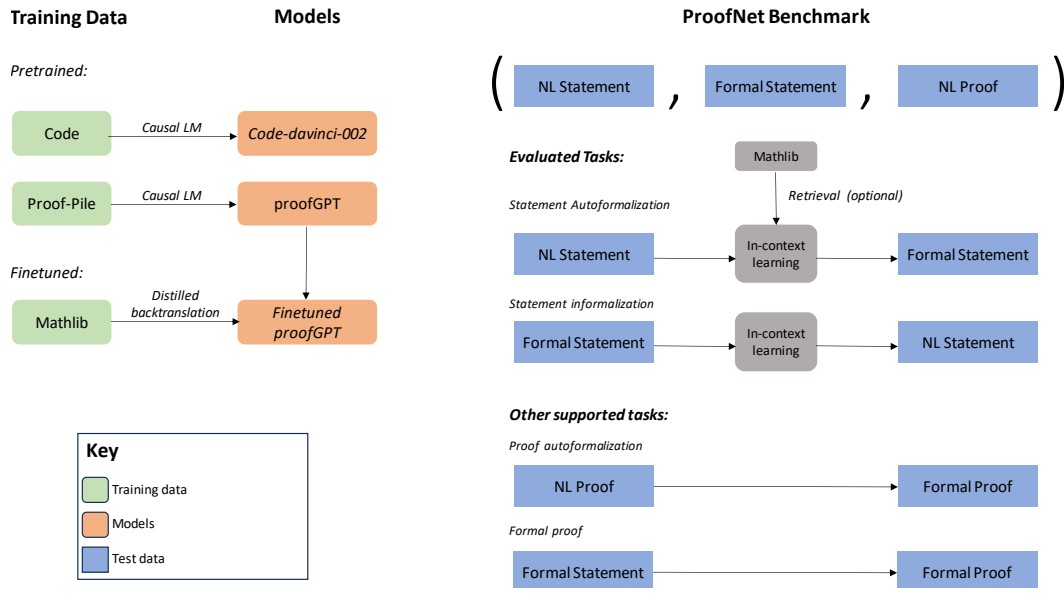

Figure 2: **Left:** We focus our evaluation on three language models. The first is the *Code-davinci-002* endpoint of the OpenAI API (Chen et al., 2021), which is pre-trained on a (proprietary) code dataset. The second is the PROOFGPT suite, which is pre-trained on the proof-pile dataset. Finally, we also finetune a PROOFGPT model using the distilled backtranslation methodology (see subsubsection 4.1.3). **Right:** Each example in the ProofNet benchmark consists of a natural language (NL) statement, a formal statement, and an NL proof. In this work, we focus our evaluation on statement autoformalization and informalization. The tasks of proof autoformalization and formal theorem proving are also supported by ProofNet.

## 2   THE ProofNet BENCHMARK

**Dataset collection**   Problems in the ProofNet benchmark are primarily drawn from exercises in popular undergraduate mathematics textbooks. For a complete list of sources, see Appendix B. For a comparison of ProofNet to other mathematical reasoning evaluations, see Appendix C

Not all textbook exercises lend themselves naturally to formalization. In particular, we only consider for inclusion in ProofNet problems meeting the following criteria:

- *Self-containment.* Problems should only depend on the results commonly taught in an undergraduate curriculum. In particular, this rules out problems that are split into multiple sequentially dependent parts, or those using nonstandard notations.

- *Naturality of formalization.* Not all kinds of mathematical problems can be naturally formalized, such as word problems, and such problems are excluded. We do not include exercises that require computing an unknown quantity. We do not include problems that depend on parts of Lean's mathlib that are relatively less mature, such as Euclidean geometry or combinatorics.

- *Low risk of train-test overlap.* Because language models are often pre-trained on large corpora mined from the internet that include mathlib, we refrain from including statements that are in mathlib or are likely to be added to mathlib in the future. In practice, this means we avoid the abstract "theory-building" style of theorems that constitute mathlib, and instead choose problems that involve applying general results to specific cases. For more insight into the stylistic differences between mathlib and ProofNet problems, see Figure 1.

Beyond the above criteria, problems were selected for broad coverage of the undergraduate curriculum and to range in difficulty from straightforward applications of the definitions to those requiring tremendous creativity. Problems statements are transcribed into LATEX and formalized by human

| Source | Size (GB) | Tokens |
|---|---|---|
| arXiv.math | 13.6 | 8.0B |
| Stack Exchanges | 0.96 | 0.3B |
| Formal math libraries | 0.14 | 59M |
| ProofWiki + Wikipedia math articles | 0.02 | 6.6M |
| Open source books | 0.015 | 6.5M |
| MATH | 0.002 | 0.9M |

Table 1: Composition of the proof-pile.

| Model | arXiv.math perplexity | proof-pile perplexity |
|---|---|---|
| *1B parameters:* | | |
| Pythia 1.4B | 3.82 | 4.12 |
| PROOFGPT 1.3B | 3.17 | 3.47 |
| *6B parameters:* | | |
| Pythia 6.9B | 3.36 | 3.62 |
| PROOFGPT 6.7B | 3.12 | 3.43 |

Table 2: Comparison of model perplexities on the test set of the arXiv subset of the proof-pile and the entire proof-pile. Documents were joined using two newline characters and perplexity was calculated with a stride equal to the model's context length, which is 2048 for all models shown.

annotators proficient in Lean. Natural language proofs are adapted from online solutions manuals, or in a few cases, written by the annotators.

**Supported tasks** As ProofNet includes parallel natural language statements, natural language proofs, and formal statements, the dataset supports the evaluation of the following distinct tasks:

- *Formal theorem proving*. Given a formal statement of a theorem, produce a formal proof.
- *Informal theorem proving*. Given an informal statement, produce an informal proof. This facilitates direct comparison between formal and informal theorem proving approaches.
- *Autoformalization and informalization of statements*. Given an informal (formal) statement, produce a corresponding formal (informal) statement.
- *Autoformalization of proofs*. Given an informal theorem statement, its informal proof, and its formal statement, produce a formal proof.

## 3 THE PROOFGPT MODELS AND THE proof-pile DATASET

In order to obtain stronger baselines on the ProofNet benchmark, we introduce the PROOFGPT language models and a text dataset named the proof-pile that these models are trained on. Many approaches to quantitative reasoning with language models depend on pre-training or fine-tuning a model on large corpora of mathematical text, which significantly boosts downstream performance (Hendrycks et al., 2021b; Polu and Sutskever, 2020; Lample et al., 2022; Lewkowycz et al., 2022). Motivated by these results, we train and open-source the PROOFGPT models at sizes of 1.3 billion and 6.7 billion parameters.[2] The PROOFGPT models are decoder-only causal language models initialized with Pythia weights (Biderman et al., 2023),[3] and then fine-tuned on the proof-pile,[4] a

---

[2]We will make these models available upon acceptance.

[3]The PROOFGPT models were not initialized from the open-sourced weights of the Pythia models, but from a development version of the suite with slightly different architecture and training hyperparameters. This is the cause of the small parameter discrepancy between a PROOFGPT and the similarly sized Pythia model. Performance of the development versions of Pythia and the open-source versions are near-identical.

[4]Data will be open-sourced upon acceptance.

corpus of unstructured mathematical text gathered from internet sources whose composition is detailed in Table 1. The proof-pile contains roughly 8.3 billion GPT-NeoX (Andonian et al., 2021) tokens. Fine-tuning was performed using the GPT-NeoX library (Andonian et al., 2021). For training hyperparameters, see Appendix A. In Table 2, we show that the PROOFGPT models outperform Pythia base models at standard mathematical reasoning tasks.

We regard the PROOFGPT model suite as inferior to the Minerva models (Lewkowycz et al., 2022) due to the fact that the PROOFGPT models are fine-tuned on an order of magnitude less mathematical text and span a smaller parameter range. However, we hope that the research community will benefit from PROOFGPT's open-source weights and dataset.

## 4    METHODOLOGY AND EXPERIMENTS

In this work, we evaluate the capabilities of pre-trained language models on autoformalizing and informalizing theorem statements. Due to the engineering challenges of implementing neural theorem proving systems in Lean, we leave an investigation of formal theorem proving and proof autoformalization to future work.

### 4.1    AUTOFORMALIZATION METHODS

We employ in-context learning with large language models as a strong baseline for the autoformalization of theorem statements (Wu et al., 2022a). Moreover, we introduce two novel methods for boosting autoformalization performance above the few-shot baseline: *prompt retrieval* and *distilled backtranslation*.

#### 4.1.1    FEW-SHOT AUTOFORMALIZATION AND INFORMALIZATION

In-context learning is a simple and powerful method for adapting language models to sequence-to-sequence tasks (Brown et al., 2020).

For our in-context baselines, we perform inference using the OpenAI API's *Code-davinci-002* endpoint (Chen et al., 2021) and the PROOFGPT 1.3B and 6.7B models. Prompts are listed are given in Appendix D.

Because there may be multiple ways to formalize the same statement in Lean and no general way to automatically verify whether two statements that are not definitionally equal have the same mathematical content, autoformalizations should be evaluated for correctness by a human expert. For similar reasons, informalizations should also be judged by human experts. In this work, model outputs are scored by the authors. Our open-source repository contains raw model outputs so that the author's judgements of correctness can be independently verified.

#### 4.1.2    PROMPT RETRIEVAL

A blessing and a curse of current language models is that few-shot learning performance is highly sensitive to the exact prompt that is used (Kojima et al., 2022). In particular, it is plausible that greater few-shot learning performance can be achieved by retrieving the few-shot examples that are most relevant to a particular question.

Following Liu et al. (2022), we implement a *prompt retrieval* procedure for statement autoformalization based on nearest neighbors search. Suppose we have a knowledge-base $\mathcal{K}$ of formal statements. First, we generate an autoformalization $\hat{y}$ of a statement $x$ using our standard in-context procedure. Then we produce dense vector representations of $\hat{y}$ and the formal statements in $\mathcal{K}$. We retrieve the $k$-nearest-neighbors of $\hat{y}$ in $\mathcal{K}$, and include them in the few-shot prompt. For the precise format of the prompt, see Appendix D.

We opt to retrieve against $\hat{y}$ instead of against $x$ because this method was significantly more performant in our preliminary experiments.

In our experiments, we create a knowledge-base $\mathcal{K}$ by taking our $y$s to be 90,530 statements from Lean mathlib and use $k = 4$. We use the OpenAI API's *embedding-ada-002* endpoint Neelakantan et al. (2022) to generate text embeddings.

| | Formalization | | | Informalization | | |
|---|---|---|---|---|---|---|
| Model | Typecheck rate | BLEU | Accuracy | Compile rate | BLEU | Accuracy |
| *Few-shot.* | | | | | | |
| PROOFGPT-1.3B | 5.9 | 8.1 | 0 | 77 | 5.1 | 4.3 |
| PROOFGPT-6.7B | 4.3 | 4.7 | 0 | 70 | 6.0 | 6.5 |
| *Code-davinci-002* | 23.7 | 25.1 | 12.9 | 100 | 13.2 | 62.3 |
| *Prompt retrieval:* | | | | | | |
| *Code-davinci-002* | 45.2 | 14.8 | 15.6 | - | - | - |
| *Dist. backtrans.* | | | | | | |
| PROOFGPT-1.3B | 19.4 | 10.7 | 3.2 | - | - | - |

Table 3: Results of few-shot learning with LLMs on formalization and informalization of ProofNet statements; all cells are percentages. In addition to reporting autoformalization accuracy, we also report *typecheck rate*, which is the proportion of a model's samples that are well-formed statements in Lean's dependent type theory. If a model simply copies a formal statement from its prompt, we do not consider that a positive sample when calculating typecheck rate. For the informalization task, we also report *compile rate*, i.e., what proportion of the model's samples produce LaTeX that compiles. The most common reason why informal generations fail to compile is that they contain Unicode characters frequently used in Lean's mathlib but not accepted by the pdflatex compiler. To calculate BLEU scores, we split on whitespace and use BLEU-4 with smoothing. Note that formalization BLEU scores being higher than informalization BLEU scores is likely because natural language contains more lexically distinct but semantically equivalent statements.

### 4.1.3 DISTILLED BACKTRANSLATION

Due to the amount of domain expert time required to collect parallel corpora of natural language and formal mathematics, scaling up parallel datasets to the point where they are useful for supervised finetuning is impractical. In the face of this limitation, to finetune models on autoformalization we draw on prior work leveraging generative models for unsupervised translation between natural languages. In particular, we use *distilled backtranslation*, a methodology inspired by Han et al. (2021a).

Distilled backtranslation proceeds as follows. Suppose we have a large language model $P_{LLM}(\cdot)$ pre-trained on monolingual data in both the source and target language, a monolingual corpus $\{Y_i\}$ in the target language. We wish to fine-tune a "student" model $P_\theta(Y|X)$ to translate a sequence $X$ in the source language to a corresponding sequence $Y$ in the target language. First, we manually construct a few-shot prompt $C$ consisting of $X|Y$ pairs. Then, we sample synthetic backtranslations $X_i \sim P_{LLM}(X|C, Y_i)$. Finally, we fine-tune $P_\theta(\cdot)$ on the synthetic pairs to predict $P(Y|X)$.

In our experiments, we fine-tune PROOFGPT-1.3B using distilled backtranslation with informal mathematics as the source language and Lean 3 theorems as the target language. We use the theorems in Lean's mathlib as the target language's monolingual corpus. We use *Code-davinci-002* as our teacher LM and proofGPT-1.3B as our student model. Fine-tuning hyperparameters are described in Appendix E

## 5 RESULTS AND DISCUSSION

### 5.1 IN-CONTEXT LEARNING

In Table 3, we present our experimental results for autoformalization and informalization of ProofNet theorem statements. Although conceptually simple and easy to implement, our *Code-davinci-002* in-context learning baseline achieves highly nontrivial performance, correctly formalizing 12.9% of theorems. The PROOFGPT models do not formalize any statements correctly, likely owing to their smaller parameter count. However, they demonstrate some signal on the typecheck

rate and BLEU metrics. Note that even generating statements that typecheck in Lean 3's strict type system is a nontrivial feat.

Informalization accuracy is much higher than formalization accuracy for all models, supporting the intuitive claim that informalization is an easier task than formalization. This result also suggests that large pre-trained language models have a strong grasp of the semantics of formal mathematics, and primarily struggle with generating lexically correct and type-correct Lean code.

We further observe that among *Code-davinci-002*'s generations that typecheck, roughly half are correct formalizations. This is consistent with our hypothesis that *Code-davinci-002* has a strong grasp of the semantics of mathematics, since the model displays high accuracy conditional on having generated valid Lean.

## 5.2 PROMPT RETRIEVAL AND DISTILLED BACKTRANSLATION

In Table 3, we additionally include autoformalization scores for the prompt retrieval and distilled backtranslation models. Applying prompt retrieval to the *Code-davinci-002* model significantly boosts performance, increasing accuracy by 2.7 points and, notably, increasing typecheck rate by 21.5 points.

Distilled backtranslation improves the autoformalization performance of the PROOFGPT 1.3B model not merely above the in-context performance of PROOFGPT 1.3B, but also above the in-context learning performance of PROOFGPT 6.7B.

**Automatic metrics** Typecheck rate correlates strongly with formalization accuracy, and we recommend that typecheck rate be used as a predictor of autoformalization performance when evaluating accuracy is too costly. The BLEU metric correlates well with performance on many NLP tasks (Papineni et al., 2002), but correlates poorly with performance code tasks (Chen et al., 2021). Our findings illustrate that just as with code, BLEU is a poor guide to formalization performance, as prompt retrieval increases *Code-davinci-002* formalization accuracy but decreases BLEU by over 10 points.

## 5.3 QUALITATIVE ANALYSIS

We ground our remaining analysis in four case studies: two that demonstrate successful formalizations produced by *Code-davinci-002*, and two that are representative of our methods' most common failure cases.

**Case study 1.** The following is a Lean formalization of exercise 4.21a in Walter Rudin's *Principles of Mathematical Analysis* produced by *Code-davinci-002*:

NL: Suppose $K$ and $F$ are disjoint sets in a metric space $X$, $K$ is compact, $F$ is closed.
Prove that there exists $\delta > 0$ such that $d(p, q) > \delta$ if $p \in K, q \in F$.

*Code-davinci-002* output:
```
theorem exercise_4_21a {X : Type*} [metric_space X] {K F : set X}
  (hK : compact_space K) (hF : is_closed F) (hKF : disjoint K F) :
  ∃ (δ : ℝ), δ > 0 ∧ ∀ (p q : X), p ∈ K → q ∈ F → dist p q ≥ δ
```

In this example, *Code-davinci-002* demonstrates strong knowledge of `mathlib` and some reasoning capability. For example, the model correctly identifies that metric scapes are denoted using a typeclass `metric_space X`, but closed sets are expressed using a predicate `is_closed`. Moreover, the model correctly renders a consequent that contains nested quantifiers. Finally, the model recognizes the equivalence between "if $p \in K$ and $q \in F$" and "$p \in K \rightarrow q \in F \rightarrow$", which demonstrates some logical reasoning ability. However, note that the formalization is not completely idiomatic, as `[compact_space K]` or `(hK : is_compact K)` would be preferred to the model's `(hK : compact_space K)`.

**Case study 2.** The following problem, exercise 6.4.2 from Michael Artin's *Algebra*, is an example of vanilla *Code-davinci-002* making a mistake that is rectified by prompt retrieval.

---

NL: Prove that no group of order $pq$, where $p$ and $q$ are prime, is simple.

---

*Code-davinci-002* output:
```
theorem exercise_6_4_2 {G : Type*} [group G] [fintype G] {p q :  ℕ}
(hp :  nat.prime p) (hq :  nat.prime q) (h :  fintype.card G = p * q) :
¬ simple_group G
```

---

*Code-davinci-002* + retrieval output:
```
theorem exercise_6_4_2 {G : Type*} [group G] [fintype G] {p q :  ℕ}
[hp :  fact(nat.prime p)] [hq :  fact(nat.prime q)]
(h :  fintype.card G = p * q):
¬ is_simple_group G
```

---

Vanilla *Code-davinci-002* makes a single lexical error: the model has correctly formalized the hypotheses of the problem, but has tried to use a nonexistent predicate `simple_group` instead of the correct `is_simple_group`. The retrieval-augmented model uses the correct declaration because one of the retrieved mathlib statements explicitly uses `is_simple_group`.

**Case study 3.** The following is an incorrect formalization of exercise 6.16 from Sheldon Axler's *Linear Algebra* produced by *Code-davinci-002*. The retrieval-augmented *Code-davinci-002* autoformalization, not shown here, is also incorrect.

---

NL: Suppose $U$ is a subspace of $V$. Prove that $U^\perp = \{0\}$ if and only if $U = V$

---

*Code-davinci-002* output:
```
theorem exercise_6_16 {K : Type*} [field K]
 {V : Type*} [add_comm_group V] [vector_space K V] {U : subspace V} :
 U.orthogonal = {0} ↔ U = ⊤
```

---

ProofNet label:
```
theorem exercise_6_16 {K : Type*} [is_R_or_C K]
 [inner_product_space K V] (U : submodule K V) :
 U.orthogonal = ⊥ ↔ U = ⊤
```

---

Here, *Code-davinci-002* makes three straightforward lexical mistakes: in mathlib, vector spaces over a field $K$ are denoted by `module K V` rather than `vector_space K V`, subspaces of a vector space by `submodule K V` rather than `subspace V`, and the trivial subspace is denoted by $\perp$ rather than `{0}`. However, the model also makes a much more significant logical error. In order for the orthogonal complement $U^\perp$ of a subspace to make sense, the space must be endowed with a *inner product space* structure rather than merely a vector space structure, which is expressed in mathlib as `inner_product_space K V`. Furthermore, inner product spaces are only defined over the real and complex fields, so one must also declare `[is_R_or_C K]`. Reliably inferring these kinds of implicit hypotheses is a major challenge for autoformalization systems.

**Case study 4: looping.** When generating formal and natural language theorem statements with PROOFGPTs, the model will often repeatedly generate the same short phrase instead of producing a plausible theorem. For example, consider the attempted formalization of exercise 10.1.13 from Michael Artin's *Algebra* generated by PROOFGPT 6.7B via in-context learning.

---

NL: An element $x$ of a ring $R$ is called nilpotent if some power of $x$ is zero.
Prove that if $x$ is nilpotent, then $1 + x$ is a unit in $R$.

---

PROOFGPT-6.7b output:
```
theorem nilpotent_of_nilpotent_of_nilpotent_of_nilpotent_of_nilpotent
nilpotent_of_nilpotent_of_nilpotent_of_nilpotent_of_nilpotent_of...
```

---

Prior work on decoding methods has shown that the likelihood of a repeated phrase increases with each repetition, and that greedy decoding generates text with higher likelihood than natural text (Holtzman et al., 2019). These two findings constitute a plausible explanation for repetitive looping if the correct autoformalization is assigned low likelihood by the model. We observe that repetitive looping does not occur with *Code-davinci-002*, suggesting that the problem may disappear with scale (although there are many other differences between our small-scale models and *Code-davinci-002*).

## 6 RELATED WORK

**Language modeling for theorem proving**    Language models have found success in theorem proving both in the natural language setting (Lewkowycz et al., 2022; Welleck et al., 2021a), and within many major ITPs such as Metamath (Polu and Sutskever, 2020), Isabelle (Jiang et al., 2022a; First et al., 2023; Mikula et al., 2023), and Lean (Han et al., 2021b; Polu et al., 2022; Yang et al., 2023). Popular benchmarks for evaluating language model-based provers are Hendrycks et al. (2021b) and Welleck et al. (2021a) for natural language, and Zheng et al. (2022) for formal.

**Autoformalization**    Recent work in autoformalization with language models was sparked by Wu et al. (2022a), which demonstrated that models can autoformalize Isabelle theorem statements via in-context learning. In Jiang et al. (2022b), the authors demonstrate a method for autoformalizing proofs in Isabelle. However, their method depends on the availibility of a performant black-box automated theorem prover, which is not available for Lean at the time of writing.

**Interactive theorem proving**    Work in formal theorem proving and autoformalization depends on libraries of formalized mathematics. This work directly depends on Lean's mathlib, but indirectly benefits from lessons learned from other proofs systems such as Coq (Bertot and Castéran, 2004), Mizar (Grabowski et al., 2010), and Isabelle (Nipkow et al., 2002).

**Unsupervised machine translation**    Because the amount of parallel formal and natural language text is negligible, autoformalization faces many of the same challenges as unsupervised machine translation (Lample et al., 2017; Conneau et al., 2018; Lample et al., 2018; Han et al., 2021a; Garcia et al., 2023). Our distilled backtranslation method is inspired by the distilled and iterated backtranslation algorithm of Han et al. (2021a). However, the authors of this work regard backtranslation as a temporary workaround and foresee that in-context learning will be enough to elicit maximal performance from a sufficiently good language model, as is now the case for unsupervised translation (Garcia et al., 2023).

## 7 CONCLUSION

We introduced ProofNet, a benchmarking consisting of parallel natural language theorem statements, natural language proofs, and formal theorem statements in Lean 3. We have shown that pre-trained large language models achieve non-trivial but far from consistent performance via in-context learning on the autoformalization of ProofNet statements. Moreover, we have proposed prompt retrieval and distilled backtranslation, two methods that improve autoformalization performance above baseline.

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

|  | Setting | |
| --- | --- | --- |
| Parameter | 1.3B | 6.7B |
| Tokens | 10.5 billion | |
| Epochs | 1.3 | |
| Training Steps | 40,000 | |
| Learning Rate Max | $2 \cdot 10^{-4}$ | $1.2 \cdot 10^{-4}$ |
| Learning Rate Min | $2 \cdot 10^{-5}$ | $1.2 \cdot 10^{-5}$ |
| Optimizer | Adam | |
| Adam Betas | $(0.9, 0.95)$ | |
| Adam Eps | $1 \cdot 10^{-8}$ | |
| Weight Decay | 0.1 | |
| LR Scheduler | Cosine w/ warm-up | |
| LR Warm-up Steps | 400 | |
| Effective Batch Size | 128 | |
| Precision | FP16 | |
| Gradient Clipping | 1.0 | |

Table 4: PROOFGPT training hyperparameters.

APPENDIX

A    PROOFGPT TRAINING

Table 4 displays hyperparameters for PROOFGPT training on the proof-pile.

B    PROBLEM SOURCES

The following is a complete list of sources ProofNet draws from:

- Analysis: Walter Rudin's *Principles of Mathematical Analysis* 3rd ed, Charles C. Pugh's *Real Mathematical Analysis* 1st ed, Elias M. Stein and Rami Shakarchi's *Complex Analysis* 1st ed.
- Linear Algebra: Sheldon Axler's *Linear Algebra Done Right* 2nd ed.
- Abstract Algebra: David S. Dummit and Richard M. Foote's *Abstract Algebra* 3rd ed, I.N. Herstein's *Abstract Algebra* 3rd ed, and Michael Artin's *Algebra* 1st ed.
- Topology: James Munkres' *Topology* 2nd ed.
- Examinations: Putnam Competition.

C    COMPARISON TO EXISTING BENCHMARKS

For a comparison of ProofNet to existing mathematical reasoning benchmarks, see Table 5.

D    PROMPTS

Prompts are viewable in the open-source repository [5] The retrieval knowledge base and the code for generating it is also available in the repository[6] We use a 12-shot prompt for *Code-davinci-002* autoformalization and informalization, and a 6-shot prompt for PROOFGPT autoformalization and informalization. We give PROOFGPT models fewer examples because of its shorter context (2048 tokens compared to 8192), we only use the last six examples when prompting PROOFGPT.

For retrieval augmented models, we use a 3-shot prompt, where each example consists of 4 reference formal statements and one NL-formal pair.

---

[5]Release pending acceptance.
[6]Release pending acceptance.

| | MATH[*] | MMLU-STEM[**] | PISA[***] | MiniF2F[†] | NaturalProofs[††] | GHOSTS [†††] | ProofNet (ours) |
|---|---|---|---|---|---|---|---|
| Contains formal? | ✗ | ✗ | ✓ | ✓ | ✗ | ✗ | ✓ |
| Contains natural language? | ✓ | ✓ | ✗ | ✓ | ✓ | ✓ | ✓ |
| Problem Level[a] | HS | HS+UG | Unrestricted | HS | UG+G | HS+UG+G | UG |
| Problem diversity[b] | Low | High | High | Low | High | High | High |
| Answer format | Numerical | Multi-choice | Text | Text | Text | Text | Text |
| Multi-task | ✗ | ✗ | ✗ | ✓ | ✗ | ✓ | ✓ |
| Proof-based task available? | ✗ | ✗ | ✓ | ✓ | ✓ | ✓ | ✓ |
| Training set? | ✓ | ✓ | ✓ | ✗ | ✓ | ✗ | ✗ |
| Validation + test size | 5000 | 3364 | 4000 | 488 | 3825 | 733 | 371 |

Table 5: A comparison of ProofNet to standard benchmarks for evaluating the mathematical capabilities of language models. [*] Hendrycks et al. (2021b). [**] Hendrycks et al. (2021a). [***] Jiang et al. (2021). [†] Zheng et al. (2022); Jiang et al. (2023). [††] Welleck et al. (2021b). [†††] Frieder et al. (2023). [a]: HS refers to "high school" UG refers to "undergraduate", and G refers to "graduate". The problem level of PISA is referred to as "unrestricted" because PISA is based on the Archive of Formal Proofs (AFP) (Isa), which is a library of formalized mathematics containing theorems at a wide variety of levels. [b]: MATH and MiniF2F are labelled as low diversity because they only contain high-school level Olympiad problems. MMLU-STEM, PISA, and ProofNet are labelled as high diversity because they covers multiple parts of the mathematics curriculum.

| Parameter | Setting |
|---|---|
| Training Steps | 20,000 |
| Learning Rate (LR) | $5 \cdot 10^{-5}$ |
| Optimizer | AdamW |
| Adam Betas | $(0.9, 0.999)$ |
| Adam Eps | $1 \cdot 10^{-8}$ |
| Weight Decay | 0.1 |
| LR Scheduler | Cosine w/ warm-up |
| LR Warm-up Steps | 2000 |
| Effective Batch Size | 24 |
| Precision | FP16 |
| Gradient Clipping | 1.0 |

Table 6: Student training hyperparameters.

# E    FINETUNING

Our fine-tuning dataset of backtranslations consists of 90,530 NL-formal pairs. Both the Pythia-1.4b and PROOFGPT-1.3B model are finetuned according to the hyperparameters above. The models evaluated in Table 3 are the minimum validation loss checkpoint, which occurs at 15,000 training steps.

