# OpenReview forum: "ProofNet: Autoformalizing and Formally Proving Undergraduate-Level Mathematics"
_ICLR.cc/2024/Conference — Submitted to ICLR 2024_

### Official Review · Reviewer_r6yD · 2023-10-26

**Soundness:** 3 good
**Presentation:** 3 good
**Contribution:** 3 good
**Rating:** 5
**Confidence:** 3

**Summary:**

This paper introduces the ProofNet benchmark (371 examples). Each example includes a formal theorem statement (Lean 3), a natural language theorem statement, and a natural language proof. The dataset supports the following tasks: (1) input: formal theorem statement, output: formal proof; (2) input: informal statement, output: informal proof; (3) input: informal (formal) statement, output formal (informal) statement; (4) input: informal statement, output: formal proof.

The authors also train and open-source ProofGPT (1.3B and 6.7B models) trained on proof-pile – 8.3B tokens of mathematical text (authors promise to open-source).

The authors run baselines (using ProofGPT with two sizes as well as code-davinci-002), including few-shot baselines. The authors also attempt two other approaches: prompt retrieval and distilled backtranslation. Informalization (formal statement converting to informal statement) accuracy is higher than formalization. Although the datasets can support many tasks (mentioned above), most discussion in this paper is on autoformalization (informal -> formal) and informalization (formal -> informal).

**Strengths:**

Each example contains the formal theorem statement, natural language statement, and natural language proof.

Theorem proving is extremely interesting and deserves attention. I was trying to investigate this direction (autoformalizing proofs, training math LLMs, analyzing accuracy of proofs -- especially awarding partial accuracy) but gave up due to a few roadblocks. I'm really glad the authors spent time on this topic.

In general, I consider this paper to be unique and definitely useful contribution to the community.

**Weaknesses:**

This benchmark is great. It’d be more interesting in the long term if there are questions that are definitely unseen (or not seen too much) in the pretraining corpus (not appearing in arxiv, lecture notes, textbooks, mathoverflow, etc.). For example, Rudin may be too popular online, appearing often in various websites. Perhaps variations of the existing questions or homework assignments (with solutions that can’t be found online)?

There doesn’t seem to be much discussion on the theorem proving results, especially automatic metrics. It’d be good to attempt to provide an evaluation metric for judging the quality of informal (or formal) proofs – especially on how to award partial effort. If that’s too difficult, discuss how different human judgments are from machine judgment (for informal or formal proofs).

Update: I also share the concern of Reviewer 4S8n's last "weakness" re: how to use the benchmark.

**Questions:**

What’s the specific decoding methods? More discussion please. Apologies if I missed it.

**Details Of Ethics Concerns:**

This paper is unique & great contribution. But I'd run into major trouble if I try to publish this paper (bc of my lab / employers). But unclear what the authors' situation is. Specifically, many textbooks (containing questions used in this benchmark) like Rudin's Analysis or the Topology or Algebra textbooks may have strict copyright/license (see appendix).

---

### Official Review · Reviewer_TFoh · 2023-10-31

**Soundness:** 3 good
**Presentation:** 2 fair
**Contribution:** 2 fair
**Rating:** 5
**Confidence:** 4

**Summary:**

In this submission, the authors provide a new benchmark, called ProofNet benchmarks, for evaluating the auto formalization abilities of Transformers. The benchmark consists of 371 examples, each a formal theorem statement in Lean 3 and a natural language theorem statement, and a natural language proof. The problems are drawn from popular undergraduate math textbooks. Additionally, the authors trained a transformer from scratch applying prompt retrieval and distilled back translation.

**Strengths:**

- The benchmark is a valuable contribution to the open source research community
- The from scratch trained models can serve as a baseline for future research
- Qualitative Analysis is interesting

**Weaknesses:**

- The comparison against Code-davinci-002 are unfair (as mentioned in the paper; due to the difference in parameter size and training)
- “we introduce two novel theorem autoformalization methods”; some of the claimed contributions in the second half of the paper are, unfortunately minor
- It is unclear if the proof-pile is a contribution of the authors or an already existing dataset; (if so citation is missing)

**Questions:**

- What was the reason for training models from scratch instead of fine-tuning a pretrained model?
- It is known (from Minerva for example) that significantly more data, a well behaved distribution and much more parameters are needed to obtain a good performance. How would the authors envision that the ProofGPT models will be of value to the research community?

---

### Official Review · Reviewer_RZYR · 2023-10-31

**Soundness:** 3 good
**Presentation:** 2 fair
**Contribution:** 3 good
**Rating:** 3
**Confidence:** 4

**Summary:**

This paper propose a new benchmark, ProofNet, for the task autoformalization and formal theorem proving. ProofNet contains 371 exercise-like math problems of undergraduate level mathematics. Two baselines are built for autoformalization on this benchmark, the in-context prompting of Code-davinci-002, and ProofGPT, a Pythia model finetuned on proof-pile, a new collection of math related training corpora. The in-context learning baseline is further improved by prompt retrieval. The ProofGPT baseline is further improved by distilled backtranslation.

**Strengths:**

1 The proposed ProofNet could be a valuable test suite for benchmarking automated theorem proving. Unlike most formal mathematical corpora that formalize mostly the fundamental math concepts and theorems, ProofNet share the same style as miniF2F that provides a collection of regular math questions that can be used in exercises and exams. It is more natural to evaluate the performance of a theorem prover with these exercise-style questions.
2 On top of the two simple baselines (in-context prompting and ProofGPT), this paper studies two followup techniques for improving their results, prompt retrieval and distilled backtranslation. Both techniques are appropriately applied in the context of autoformalization. Experiments also demonstrate the effectiveness of these two methods.

**Weaknesses:**

1 The presentation of the paper could be improved. Especially, it would be helpful to summarize of the main contributions of this paper, like ProofNet, ProofGPT, and baseline results.
2 Besides autoformalization, automated theorem proving obtains more attentions. Due to the missing of the baseline results of theorem proving on ProofGPT, it is unclear how difficult this benchmark is.

**Questions:**

1 Do you have a general idea on how difficult the ProofNet problems are compared to miniF2F in terms of theorem proving?
2 Code-davici-002 is not the best pretrained LLM. To get a stronger baseline, it would be helpful to have the results of GPT4 on the ProofNet benchmark.

**Details Of Ethics Concerns:**

No ethics concerns

---

### Official Review · Reviewer_4S8n · 2023-11-01

**Soundness:** 3 good
**Presentation:** 3 good
**Contribution:** 2 fair
**Rating:** 5
**Confidence:** 4

**Summary:**

The authors proposed a benchmark for autoformalization and formal proving of undergraduate-level mathematics in Lean. The authors also reported the performance of three baseline approaches (e.g., few-shot learning, prompt retrieval and distilled backtranslation) to autoformalization on the benchmark.

**Strengths:**

- The benchmark covers a good variety of undergraduate-level math problems.
- Some ideas in the baselines approaches are inspiring. For example, prompt retrieval against the target instead of the input.
- Good qualitative case studies. These are actually a must since there is no general way to test whether two statements are definitionally equal and the correctness has to be evaluated by experts. As far as I know, there haven’t been many papers on autoformalization that carry out such studies.
- The paper is clearly written and easy to follow.

**Weaknesses:**

- The authors claimed autoformalization of proofs as a supported task,  but there doesn’t seem to be any evaluation on that. Since ProofNet does not contain ground truth of formal proofs, how would one autoformalization an NL proof? My impression is that at least a few formal proofs are required to perform any of the baseline approaches.
- Although there are some minor adaptation to existing techniques, the learning approach seems mostly standard.
- The intended use of the benchmark looks a bit confusing. If the quality of (in)autoformalization has to be evaluated eventually by human experts, then the labels in ProofNet do not seem to provide too much. After all, matching strings (or even computing BLEU scores) doesn’t make sense in the context of math. If the benchmark is supposed to be used as examples for few-shot learning, then I would suggest include at least 2-3 different formalizations of a problem to help with generalizability.

**Questions:**

See above.

---

### Meta-Review · Area_Chair_dqLr · 2023-12-12

**Metareview:**

This paper proposed a new benchmark for autoformalization and formal theorem proving. Reviewers appreciate the variety of the problems and the good qualitative case studies, but think the novelty of the methods and the evaluation are relatively limited. In addition to autoformalization, results on theorem proving using SoTA LLMs are also expected. AC agrees this work needs further improvement to be more interesting given recent advances on LLMs and autoformalization/theorem proving.

**Justification For Why Not Higher Score:**

The novelty of the methods and the evaluation are relatively limited.

**Justification For Why Not Lower Score:**

N/A

---

### Decision · Program_Chairs · 2024-01-16

Reject